# Assembly of a G-Quadruplex Repair Complex by the FANCJ DNA Helicase and the REV1 Polymerase

**DOI:** 10.3390/genes11010005

**Published:** 2019-12-19

**Authors:** Kaitlin Lowran, Laura Campbell, Phillip Popp, Colin G. Wu

**Affiliations:** 1Department of Chemistry, Oakland University, Rochester, MI 48309, USA; kalowran@oakland.edu (K.L.); lcampbell2@oakland.edu (L.C.); 2Case Western Reserve University, Cleveland, OH 44106, USA; pgp19@case.edu

**Keywords:** G-quadruplexes (G4s), 8-oxoguanine (8oxoG), helicases, FANCJ, REV1, PCNA, DNA repair, fluorescence spectroscopy, biolayer interferometry, circular dichroism

## Abstract

The FANCJ helicase unfolds G-quadruplexes (G4s) in human cells to support DNA replication. This action is coupled to the recruitment of REV1 polymerase to synthesize DNA across from a guanine template. The precise mechanisms of these reactions remain unclear. While FANCJ binds to G4s with an AKKQ motif, it is not known whether this site recognizes damaged G4 structures. FANCJ also has a PIP-like (PCNA Interacting Protein) region that may recruit REV1 to G4s either directly or through interactions mediated by PCNA protein. In this work, we measured the affinities of a FANCJ AKKQ peptide for G4s formed by (TTAGGG)_4_ and (GGGT)_4_ using fluorescence spectroscopy and biolayer interferometry (BLI). The effects of 8-oxoguanine (8oxoG) on these interactions were tested at different positions. BLI assays were then performed with a FANCJ PIP to examine its recruitment of REV1 and PCNA. FANCJ AKKQ bound tightly to a TTA loop and was sequestered away from the 8oxoG. Reducing the loop length between guanine tetrads increased the affinity of the peptide for 8oxoG4s. FANCJ PIP targeted both REV1 and PCNA but favored interactions with the REV1 polymerase. The impact of these results on the remodeling of damaged G4 DNA is discussed herein.

## 1. Introduction

### 1.1. What are G-Quadruplexes?

G-quadruplexes (or G4s) are stable secondary structures formed by guanine-rich nucleic acids. Single-stranded (ss) DNA sequences possessing four runs of three or more successive guanine bases can fold into G4s by Hoogsteen hydrogen bonding [1]. Their interactions are further stabilized by potassium or sodium cations. Depending on solution conditions and the intervening loop sequences, G4s can adopt parallel, antiparallel, or hybrid conformations [2,3]. These spatial arrangements have distinct optical properties that can be distinguished by circular dichroism (CD) spectroscopy [4]. Due to their inherent stability, the accumulation of G4s is toxic to cells, as it disrupts essential biological processes such as DNA replication and repair, RNA transcription, as well as mRNA translation [5]. However, by regulating where and when G4s can form, their presence can provide additional means to control gene expression and protein synthesis [6,7]. G4s are therefore attractive therapeutic targets for genetic diseases including cancer, heart failure, and Fanconi anemia [8,9,10,11,12,13,14]. Recent studies have shown that G4s are ubiquitous throughout the human genome with over 400,000 unique G4-forming DNA sequences [15,16]. A growing number of helicases and polymerases have emerged as key enzymes that unfold G4s in human cells [17,18,19,20,21,22,23,24]. Their activities are needed at G4s in a timely manner not only to maintain genetic stability, but also support cellular functions.

### 1.2. How Are G4s Removed in Cells?

Helicases and polymerases are motor proteins that couple the chemical energy from nucleotide binding and hydrolysis to do mechanical work [25,26,27]. Energy is consumed to facilitate translocation along nucleic acids and the removal of secondary structures within the lattice. Helicases are best known for their unwinding activity, which separates double-stranded (ds) DNA or dsRNA to form transient single-stranded (ss) intermediates [28,29]. Although polymerases function mainly in nucleic acid synthesis, some can unwind short duplexes with their binding free energy alone [30,31]. Similarly, the RecBCD and NS3 helicases can melt out several base pairs upon interacting with their targets in the absence of ATP [32,33]. Because helicases and polymerases are potent remodelers of nucleic acids, most enzymes exhibit robust G4-binding and/or unfolding activity in vitro [1]. Several have also been shown to act on G4s in vivo based on genetic and cell-based evidence. These include the XPD-family FANCJ and RTEL1 helicases [17,20], the RecQ-family WRN and BLM helicases [34,35], the Pif1 helicase [36], the REV1 DNA repair polymerase [37], and others reviewed here [1,38]. Defects in these enzymes stall replication and repair at G4 DNA. The molecular basis by which they are recruited to G4s are becoming clearer. REV1 has a high affinity for G4-containing sequences [18]. RecQ helicases have a conserved domain that binds directly to G4s [39]. The RHAU (or DHX36) helicase recognizes G4 RNA through a special motif at the protein N-terminus [40]. In solution, this site adopts an α-helix that is followed by an AKKQ loop [41]. It is thought that the helix stacks on top of a guanine tetrad while lysine residues within the AKKQ are anchored into the RNA backbone. FANCJ possesses an AKKQ motif that is flanked by a comparable G4-targeting peptide [42]. Enzymes participating in G4-maintenance have developed strategies to detect G4s in cells; their motor activities then take over to unfold these structures. One question remains a major barrier to research progress—how are damaged G4s processed within the cell? While the same helicases and polymerases may be involved, the presence of damage within a G4 presents an additional challenge for the enzymes to overcome. Guanine bases are vulnerable to reactive oxygen species that can convert them into 8-oxogunanines (8oxoGs). 8oxoG-modified G4s (8oxoG4s) can still fold into stable structures [43,44]. This work will use FANCJ as a model system to examine the molecular recognition of damaged G4 DNA. 

### 1.3. What Is the Role of FANCJ in G4 Repair?

The FANCJ helicase participates in the repair of interstrand crosslinks in human cells. Mutations in FANCJ or related proteins of this pathway can lead to Fanconi anemia, a rare disease marked by progressive bone marrow failure and increased susceptibility to cancer [45,46,47,48]. At least 23 Fanconi anemia complementation (FANC) groups have been identified that function in the initial binding of the lesion, incision of the two DNA strands, or resolution of the newly generated dsDNA break [48]. Not surprisingly, many FANC proteins are components of the nucleotide excision repair and homologous recombination machinery. FANCJ coordinates the repair of DNA intermediates that arise from these reactions [42,49,50]. It unfolds G4s in vitro and this activity supports DNA replication through G4-forming regions [12,17]. FANCJ binds to G4s with a modular insertion that is not found in other XPD-family members [42]. The unrelated RHAU RNA helicase shares this G4-binding site, but it is not known if FANCJ can function at 8oxoG4s [40,41]. In a current model, FANCJ repeatedly unfolds and refolds G4s near a stalled replication fork to keep the strand clear for a repair polymerase [42]. The REV1 translesion DNA synthesis polymerase has been suggested for this role because it prefers to incorporate cytosine across from any templated base [51]. REV1 also has a higher affinity for G4-containing DNA and disrupts its structure upon loading [18]. Repair polymerases are commonly recruited to damaged DNA via interactions with the PCNA sliding clamp [52]. Their contacts are mediated through a PCNA Interacting Peptide (or PIP) motif [53]. PIP-boxes have loosely defined consensus residues. Diverging PIP-like sequences can bind to other repair proteins including REV1 [54,55,56]. Because FANCJ has a potential PIP-motif based on its primary amino acid sequence, we hypothesize that this site can function either as a canonical PIP that recruits REV1 through interactions with PCNA, or as an REV1 Interacting Region (RIR) that brings the polymerase directly to G4s [42,54,55]. To test this, we have developed a biolayer interferometry assay (BLI) to monitor the molecular interactions between the FANCJ PIP, REV1, and PCNA. We have also validated this approach using fluorescence spectroscopy to probe the binding of FANCJ AKKQ to 8oxoG4s. 

## 2. Materials and Methods

### 2.1. Buffers and Reagents

All buffers and reagents were prepared with reagent-grade chemicals and double distilled water that was further purified with a Smart2Pure 6 UV/UF system (ThermoFisher, Waltham, MA, USA). Solutions were sterilized through a 0.22 micron PES filter. Experiments were performed at 25 °C in Buffer H (20 mM HEPES pH 7.5, 150 mM KCl, 5 mM TCEP, and 5% (*v/v*) glycerol) unless otherwise specified.

### 2.2. FANCJ Peptides and DNA Oligos

FANCJ peptides were synthesized by Genscript (Piscataway, NJ, USA). Lyophilized powders were dissolved in Buffer H, and peptide concentrations were determined using a NanoDrop One UV-Vis microvolume spectrophotometer (ThermoFisher) and the molar extinction coefficients (ε_280_) below (Table 1). Oligodeoxyribonucleotides were purchased from Integrated DNA Technologies (IDT, Coralville, IA, USA). DNA oligos were prepared similarly in Buffer H, and their stock concentrations were determined spectrophotometrically with the following molar extinction coefficients (ε_260_). Peptide samples were aliquoted, frozen, and stored at −20 °C until use. DNA substrates were stored at 4 °C. Prior to use, G4 substrates were heated to 95 °C for ten minutes followed by cooling to 25 °C over two hours to promote G4-formation.

### 2.3. Recombinant Proteins

The pEB022 plasmid used to produce full-length human PCNA protein was kindly provided by Dr. M. Todd Washington (University of Iowa). A codon-optimized construct (VB170525-1265 nra) used to express human REV1 CTD (amino acid residues 1157 to 1251) in *E. coli* was synthesized by VectorBuilder Inc. (Chicago, IL, USA). Plasmids were transformed into NiCo21(DE3) competent cells from New England Biolabs (Ipswich, MA, USA). Single bacteria colonies were selected on LB agar plates that were supplemented with 50 µg/mL of kanamycin (PCNA) or carbenicillin (REV1 CTD) to inoculate starter cultures in LB broth. Overnight growths were diluted 1:50 into 4 × 1L LB (37 °C, 250 rpm). Protein expression was induced by adding IPTG to 1 mM when the optical density at 600 nm reached 0.4 absorbance units. Cells were cultured overnight at 16 °C. They were harvested 16 h post induction by centrifugation (6000 *× g* at 4 °C) for 60 min, and the pellets were resuspended in 20 mM NaPi (pH 7.5), 300 mM NaCl, 30 mM imidazole, 1 mM DTT, 5% (*v/v*) glycerol, 1% NP-40, and 1 mM PMSF.

*E. coli* resuspensions were sonicated on ice at 45% amplitude using a Fisher 505 dismembrator (Fisher Scientific, Hampton, NH, USA) equipped with a 12 mm titanium probe. After a sonication time of 15 min (15 s on, 45 s off), lysates were clarified by centrifugation (45,000× *g* at 4 °C) for 90 min. Supernatant samples were filtered through a 0.22 micron PES membrane prior to liquid chromatography. 

All protein purification steps were conducted within a refrigerated cabinet maintained at 4 °C. Filtrates were loaded onto a 20 mL IMAC column (GE Healthcare, Chicago, IL, USA) that was charged with Ni^2+^ and equilibrated in 20 mM NaPi (pH 7.5), 300 mM NaCl, 30 mM imidazole, 1 mM DTT, and 5% (*v/v*) glycerol using an AKTA Start FPLC (GE Healthcare). The column was then washed with 200 mL of reduced salt buffer (20 mM NaPi (pH 7.5), 30 mM NaCl, 30 mM imidazole, 1 mM DTT, and 5% (*v/v*) glycerol). PCNA or REV1 CTD was eluted with 20 mM NaPi (pH 7.5), 30 mM NaCl, 1 M imidazole, 1 mM DTT, and 5% (*v/v*) glycerol over a 200 mL linear gradient. Fractions containing PCNA or REV1 CTD were identified by SDS-PAGE, pooled, and loaded onto a 5 mL HiTrap Heparin column (GE Healthcare) that was equilibrated in 20 mM HEPES (pH 7.5), 30 mM NaCl, 5 mM TCEP, and 5% (*v/v*) glycerol using an Azura Bio LC50 FPLC (Knauer, Berlin, Germany). The column was washed with 100 mL of the equilibration buffer to remove trace contaminants. PCNA or REV1 CTD was eluted off the column with 20 mM HEPES (pH 7.5), 1 M NaCl, 5 mM TCEP, and 5% (*v/v*) glycerol over a 50 mL linear gradient. Protein purity was assessed by SDS-PAGE. Samples were dialyzed into Buffer H with 20% (*v/v*) glycerol, aliquoted, and frozen in liquid nitrogen. Purified recombinant proteins were stored at –80 °C until use for up to 6 months. 

### 2.4. Fluorescence Spectroscopy

Equilibrium binding experiments were performed on a Cary Eclipse Fluorescence Spectrophotometer (Agilent Technologies, Santa Clara, CA, USA). Samples were maintained at 25 °C with a PCB 1500 water peltier system (Agilent). FANCJ peptide (5 µM) was placed in a microcuvette. Fluorescence spectra were collected from 300 nm to 450 nm (λ_excitation_ = 280 nm) with slit widths set to 5 nm. A scan of Buffer H was used as a baseline. DNA substrate was titrated into the cuvette, and each mixture was equilibrated for 3 min prior to data collection. The extent of fluorescence quenching observed (ΔF_obs_) was calculated using the following equation, where F_0_ was the total fluorescence signal from the FANCJ peptide alone, and F*_i_* was the total fluorescence intensity upon the *i*th addition of DNA:(1)$$\Delta F_{obs}=\frac{F_{0}-F_{i}}{F_{0}}$$

Binding isotherms were generated by plotting ΔF_obs_ versus the total DNA concentration after correcting for sample dilution. The resulting curves were fit to a 1:1 interaction model using Scientist 3.0 software (Micromath, St. Louis, MO, USA) based on the following formulas, where *A* was the amplitude of fluorescence quenching and *K* was the equilibrium association constant. *D_f_* and *D_t_* were the free and total concentration of DNA, while *P_f_* and *P_t_* described the free and total peptide concentration. Fluorescence titrations were performed in triplicate. The reported values were determined from the average and 95% confidence interval of the three independent data-sets.
(2)$$\;\Delta F_{obs}=A\left(\frac{KD_{f}}{1+KD_{f}}\right)$$
(3)$$D_{t}=D_{f}\left(1+KP_{f}\right)$$
(4)$$\;P_{t}=P_{f}\left(1+KD_{f}\right)$$

### 2.5. Circular Dichroism (CD)

CD measurements were collected on a JASCO J-810 spectropolarimeter (JASCO Inc., Easton, MD, USA) equipped with a PTC-423S peltier system. G4 DNA samples were dialyzed into 20 mM Boric Acid pH 7.5, 150 mM KCl (or NaCl). CD spectra were collected from 320 nm to 220 nm at 25 °C. Five traces were obtained for each DNA substrate, and a reference scan of the buffer alone was subtracted from the averaged CD signal.

### 2.6. Biolayer Interferometry (BLI)

BLI assays were performed on a BLItz instrument (Pall Fortébio, Fremont, CA, USA). Streptavidin-coated biosensors were purchased from Pall Fortébio (Lot 1906281). Biosensors were hydrated in Buffer H for 10 min, and the baseline optical interference was recorded. A total of 150 nM of biotinylated G4 DNA or FANCJ PIP peptide was loaded onto the sensors over 2 min, after which they were washed with Buffer H. Binding reactions were initiated by placing the biosensors in analyte solutions at the indicated concentrations. After their interactions had reached an equilibrium, the sensors were returned to buffer to monitor dissociation of the complexes. Experiments were repeated in triplicate. BLI binding isotherms were generated by plotting the plateau values of the association phase versus total analyte concentration. These curves were fit to a 1:1 interaction model, as described above. The parameters reported were obtained from the average and 95% confidence interval of the three data sets. 

### 2.7. Data Repository 

All raw and processed data files, protocols, and mathematical models used in this work can be accessed freely on the Open Science Framework.

## 3. Results

### 3.1. FANCJ AKKQ Binds to G4 DNA

To test whether the FANCJ AKKQ motif alone can target G4 structures, we used fluorescence spectroscopy to monitor the binding interactions between an FANCJ peptide and a G4 formed by the human telomeric sequence (TTAGGG)_4_. A tryptophan residue was substituted for tyrosine at the C-terminal end of SPEKTTLAAKLSAKKQASIW (FANCJ amino acids 128–147) to serve as a reporter (Figure 1A). The free peptide produced a fluorescence peak from 300 to 450 nm; equilibrium titrations of (TTAGGG)_4_ quenched this signal with increasing DNA concentration (Figure 1B). The extent of fluorescence quenching was used to generate a binding isotherm (Figure 1C). Nonlinear least squares analysis of this curve to a single-site interaction model resulted in an affinity value of K = 1.6 ± 0.7 × 10^6^ M^−1^ (20 mM HEPES pH 7.5, 150 mM KCl, 5 mM TCEP, and 5% (*v/v*) glycerol, 25 °C). We next examined how the incorporation of 8-oxoguanine affected the FANCJ AKKQ-G4 interactions. 8oxoG4 substrates with a damaged base at the first (8oxo1) or fifth (8oxo5) guanine position were used for fluorescence titrations. These sites were of interest because previous experiments showed they reduced the thermal stability of the human telomeric quadruplex modestly (ΔT_m,8oxo1_ = −12.3 °C) or severely (ΔT_m,8oxo5_ = –30.5 °C) [44,57]. The FANCJ AKKQ peptide bound to both 8oxoG4s with similar affinities as the native substrate, with K_8oxo1_ = 1.3 ± 0.4 × 10^6^ M^−1^ and K_8oxo5_ = 1.4 ± 0.5 × 10 ^6^ M^−1^ (Figure 1D). However, all three DNA sequences adopted distinct struct folds. CD spectroscopy confirmed that (TTAGGG)_4_ folded into a hybrid conformation, as expected, with a CD peak at ~290 nm, a shoulder at ~260 nm, and a trough near 240 nm (Figure 1E) [4]. The 8oxo1 and 8oxo5 modifications perturbed the G4 structure and shifted its characteristic CD profile (see also Appendix A). These sites were located within different tetrads of the G4 DNA (Figure 1F). The equilibrium binding and CD results suggested that the FANCJ AKKQ targeted the TTA loops of (TTAGGG)_4_, since 8-oxoguanine altered the hybrid G4 configuration but had a negligible effect on peptide binding.

We therefore hypothesized that a G4 substrate with shorter connecting loops between the quartets would allow FANCJ AKKQ to better “sense” DNA damage. To test this, we monitored the binding of the peptide and a quadruplex formed by (GGGT)_4_ with 8-oxoguanine at the corresponding 8oxo1 and 8oxo5 positions. This sequence was chosen because the single thymidine spacers produced a compact parallel G4 with high thermal stability (T_m_ = ~85 °C) [58]. FANCJ AKKQ bound to (GGGT)_4_ with lowered affinity K = 2.7 ± 0.4 × 10 ^5^ M^−1^ compared to (TTAGGG)_4_. The presence of oxidative damage, however, increased the equilibrium constant values ~1.5-3 fold, with K_8oxo1_ = 7.7 ± 0.7 × 10^5^ M^−1^ and K_8oxo5_ = 4.4 ± 0.6 × 10 ^5^ M^−1^ (Figure 2A). CD analysis of (GGGT)_4_ indicated the formation of a parallel G4, which had a CD maximum at ~265 nm and a minimum at ~245 nm [4]. The incorporation of 8-oxoguanine did not affect G4-folding (Figure 2B). All three DNA sequences retained a parallel conformation. These results were consistent with our working model. The thymidine handles on (GGGT)_4_ limited contacts available to the peptide and decreased its affinity for the G4. This also freed FANCJ AKKQ to bind directly to the damaged DNA sites within the 8oxo1 and 8oxo5 substrates. A summary of the equilibrium binding parameters is provided in Table 2, below.

### 3.2. An Optical Biosensor for AKKQ-G4 Binding 

We next validated our fluorescence spectroscopy results by biolayer interferometry (BLI). This technique measures the optical thickness of a biosensor surface, which can be used to examine the association and dissociation reactions of macromolecules [59,60,61]. In BLI, an incident white light passes through a biosensor, and the interference pattern of the reflected light is directly proportional to the thickness of the sensor tip (Figure 3A). When a molecule of interest is bound to the surface, the increased distance traveled by the incident light before it is reflected produces a wavelength shift (Δλ) in the optical interference. Hence, when a binding partner for the immobilized target is added, the formation of their complex will further increase Δλ or vice versa upon their dissociation. In our studies, biotinylated human telomeric G4 DNA was placed on a streptavidin-coated BLI biosensor (Figure 3B). The optical interference pattern of the sensor alone was measured in Buffer H as an initial baseline (phase 1). DNA substrate was loaded on the sensor tip, and an increase in Δλ was observed (phase 2). The sensor was then washed with Buffer H to establish a new baseline signal and to confirm that the G4 remained bound onto the surface (phase 3). FANCJ AKKQ was introduced to initiate the association reaction (phase 4). After a binding equilibrium was reached, the sensor was placed in Buffer H to monitor their dissociation (phase 5). BLI experiments were performed as a function of FANCJ AKKQ concentration, and the resulting time-courses in Figure 3C were truncated to show only the association and dissociation phases (full traces are provided in Appendix A). The plateau Δλ values at the end of the binding phase were plotted against peptide concentration to generate an isotherm; nonlinear least squares analysis of this curve to a 1:1 interaction model resulted in an equilibrium constant of K = 9.4 ± 1.1 × 10^4^ M^−1^ (Figure 3D). BLI experiments with biotinylated 8oxoG4s showed a small increase in affinity in the presence of DNA damage, with K_8oxo1_ = 2.3 ± 0.8 × 10^5^ M^−1^ and K_8oxo5_ = 1.8 ± 0.2 × 10^5^ M^−1^. These binding constants were about one order of magnitude weaker compared to the fluorescence measurements. Since the DNA was immobilized for BLI, its surface orientation on the sensor may have occluded peptide binding. This would reduce the overall affinity but enable FANCJ to interact directly with the damaged base in the 8oxo1 and 8oxo5 G4 substrates. BLI results are summarized in Table 3 below.

### 3.3. FANCJ PIP Recruits REV1 and PCNA

It was suggested that FANCJ possessed a PCNA-interaction peptide (PIP) within amino acids 1001–1017 (SWSSFNSLGQYFTGKIP), based on the underlined consensus residues [42]. PIP-containing proteins bind directly to PCNA to coordinate DNA replication and repair [53,62,63]. However, it is now evident that noncanonical PIP-boxes can serve as hotspots to recruit other proteins, including the REV1 translesion synthesis polymerase and the MLH1 mismatch repair protein [54,55,56,64]. To test if the PIP-like motif in FANCJ functions as a PIP or an REV1-interacting region (RIR), we compared how this peptide bound to PCNA versus REV1 by BLI. Recombinant human PCNA protein and the C-terminal domain (CTD) of human REV1 were purified from *E. coli* (Appendix A) [54]. Biotinylated FANCJ PIP was immobilized on a streptavidin-coated biosensor and experiments were performed as described above. As seen in the truncated time-courses, REV1 CTD produced a larger binding signal than PCNA at similar molar concentrations of protein (Figure 4A). Because Δλ is related to the thickness of the biolayer, this value reports the accumulation of mass on the sensor tip. Hence, our BLI results indicated that more REV1 CTD molecules bound to the FANCJ PIP than PCNA. This difference is further magnified as REV1 CTD has a molecular mass of ~12 kDa, while a human PCNA trimer is about seven times its size, at ~86 kDa [65,66]. Next, we tested the effects of a FANCJ PIP AA peptide on its interactions with REV1 CTD and PCNA. The consecutive consensus aromatic residues (YF) within the potential PIP-motif were replaced with alanine amino acids (AA). To our surprise, BLI time-courses of FANCJ PIP AA also showed that more REV1 CTD bound to the mutant peptide versus PCNA (Figure 4B). In fact, an overlay of the trajectories revealed that PIP AA produced larger binding signals compared to the wild-type PIP sequence for REV1 CTD (Figure 4C) and for PCNA (Figure 4D). BLI experiments with PIP and PIP AA were repeated as a function of REV1 CTD and PCNA concentration to measure their corresponding equilibrium constant values. The PIP AA mutation resulted in a small increase in affinity for REV1 CTD (Appendix A) but the binding of PCNA was weakened twofold (Appendix A). Our results suggest that the PIP sequence in FANCJ has the biochemical properties of a canonical PIP-box that binds to PCNA, but it functions better as an RIR-motif that interacts with the REV1 polymerase.

## 4. Discussion

Our fluorescence spectroscopy and BLI results indicate that the FANCJ AKKQ peptide alone can bind to G4s and 8oxoG4s. Hence, this site may be crucial for the molecular recognition of damaged G4s inside the cell. This is consistent with a previous study, in which K141/142A mutations of FANCJ AKKQ abrogated G4 binding completely [42]. Structural determinants of the interactions between FANCJ AKKQ and 8oxoG4s are currently being tested in our laboratory. We initially predicted that the incorporation of 8oxoG would destabilize G4s and that the AKKQ peptide may have a greater affinity for these structures. To our surprise, the CD measurements and binding data led to a different result. The human telomeric G4 (TTAGGG)_4_ adopted a hybrid conformation in KCl-containing buffer, and 8oxoG modifications at the 8oxo1 and 8oxo5 positions disrupted G4-folding as anticipated [44]; however, FANCJ AKKQ had the same affinity for 8oxoG4s as the native substrate. It was observed also by NMR that 8oxoG within the human telomeric G4 converted guanine glycosidic angles from *anti* to *syn* and changed the G4 configuration [43]. So how can FANCJ bind to an undamaged G4 with the same affinity as 8oxo1/5 when all three substrates folded into distinct forms? We reasoned that FANCJ AKKQ bound specifically to the TTA loop of the (TTAGGG)_4_ sequence, which would explain why the peptide was insensitive to the presence of 8oxoG in the rest of the quadruplex. Indeed, CD and binding results with a G4 formed by (GGGT)_4_ were consistent with this model. By shortening the loops that connect adjacent guanine tetrads to a single thymidine, the physical contacts available to FANCJ were limited, which reduced its overall affinity for this substrate. However, because AKKQ was no longer sequestered by the loop, the peptide could target the damaged base and we observed higher equilibrium constant values for FANCJ AKKQ binding to 8oxo1/5 for (GGGT)_4_ (Figure 5). 

Classical PIP-boxes bind to PCNA, which is a strategy used to recruit repair polymerases to damaged DNA sites [53]. Because PCNA forms a trimeric ring, it can carry up to three different repair enzymes through PIP interactions as a molecular tool belt [67]. This would enable the complex to switch polymerases in order to bypass damage-specific lesions. It has been shown that some PIP-like sequences can serve as hotspots to bind directly to other repair proteins without the need for a PCNA hub [54,68]. FANCJ possesses a potential PIP-motif based on its primary amino acid sequence [42,56]. Our BLI analysis showed that the FANCJ PIP bound to REV1 CTD and to PCNA. Interactions with REV1 CTD, however, produced a stronger binding signal than PCNA, suggesting that the formation of a PIP-CTD complex was more favorable. Interestingly, the PIP AA mutant reduced the binding constant for PCNA by half but resulted in a small increase in affinity for REV1 CTD. Corresponding double alanine substitutions within the PIP motifs of Polη and Polι have eliminated PCNA binding entirely [69]. Since FANCJ PIP AA still retained residual interactions with PCNA, it likely does not function as a canonical PIP-box. Similar mutations in RIRs have either reduced or prevented binding of REV1 [54,68]. Taken together, the FANCJ PIP has biochemical properties of PIPs and RIRs, but the consecutive aromatic amino acids within the broadly defined PIP consensus sequence are disposable for protein binding. It is possible that protein-protein interactions are mediated by an upstream phenylalanine residue (shown in red) of the site (SWSS*F*NSLGQYFTGKIP), as seen in some APIM motifs [70]. Having a PIP-box that can bind to both PCNA and REV1 may indicate that FANCJ forms a ternary complex with both proteins to process G4s. Our current research focus is using BLI to pre-form a PIP-PCNA or PIP-CTD complex, and then examining the binding interactions of the third partner. We also plan on recapitulating the G4 and protein binding activities presented here, using full-length proteins in order to reconstitute a functional G4-repair complex.

## Figures and Tables

**Figure 1 genes-11-00005-f001:**
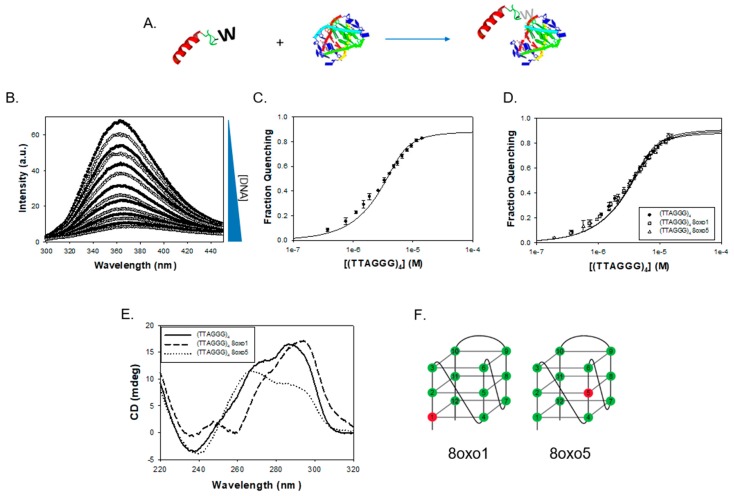
(**A**) Fluorescence spectroscopy was used to study FANCJ AKKQ-G4 binding. A tryptophan residue was incorporated into the AKKQ peptide as a fluorescence probe. (**B**) The peptide produced a large fluorescence peak alone in Buffer H (20 mM HEPES pH 7.5, 150 mM KCl, 5 mM TCEP, and 5% (*v/v*) glycerol, 25 °C). This signal was quenched upon titration of the human telomeric G4 (TTAGGG)_4_. (**C**) The extent of fluorescence quenching was plotted against total DNA concentration to generate a binding isotherm. Nonlinear least squares analysis of this curve to a 1:1 binding model resulted in an affinity value of K = 1.6 ± 0.7 × 10^6^ M^−1^. (**D**) Experiments were repeated with 8oxoG-modified substrates at the first (8oxo1) or fifth (8oxo5) guanine position. FANCJ AKKQ bound to the damaged G4s with similar affinities, with K_8oxo1_ = 1.3 ± 0.4 × 10^6^ M^−1^ and K_8oxo5_ = 1.4 ± 0.5 × 10^6^ M^−1^. (**E**) Circular dichroism spectra of the native (TTAGGG)_4_ confirmed formation of a hybrid G4 structure (solid line). The 8oxo1 (dashed line) and 8oxo5 (dotted dash) substrates disrupted G4-folding. (**F**) Schematic representation of a hybrid G4, formed by (TTAGGG)_4_. The 12 guanines within the sequence are numbered, and the damaged DNA sites of interest are highlighted in red.

**Figure 2 genes-11-00005-f002:**
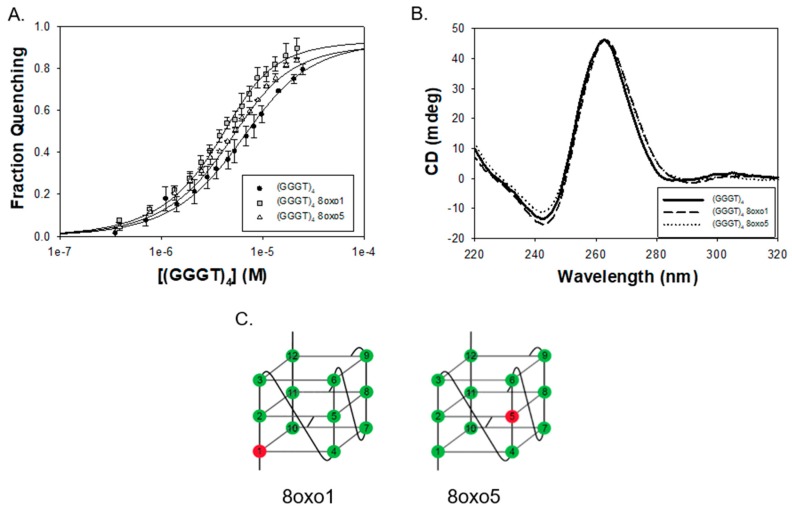
(**A**) Fluorescence spectroscopy results of FANCJ AKKQ binding to (GGGT)_4_ and substrates with 8oxoG at the first (8oxo1) or fifth (8oxo5) guanine position. Nonlinear least squares of the binding isotherms to a one-site interaction model resulted in K = 2.7 ± 0.4 × 10^5^ M^−1^, K_8oxo1_ = 7.7 ± 0.7 × 10^5^ M^−1^, and K_8oxo5_ = 4.4 ± 0.6 × 10^5^ M^−1^. (**B**) Circular dichroism spectra of the undamaged (GGGT)_4_ indicated formation of a parallel G4 structure (solid line). Incorporation of 8oxoG at the 8oxo1 (dashed line) and 8oxo5 (dotted dash) sites had no effect on G4-folding. (**C**) Cartoon diagram of a parallel G4 formed by (GGGT)_4_. The guanines within the sequence are labeled, and the damaged DNA sites of interest in this work are shown in red.

**Figure 3 genes-11-00005-f003:**
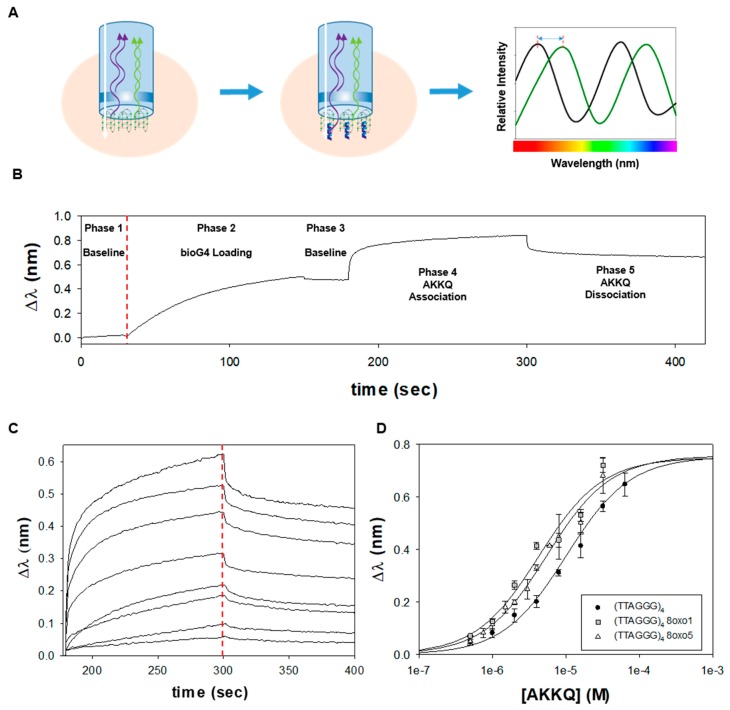
(**A**) Biolayer interferometry (BLI) overview. BLI measures the thickness of an optical surface. When an incident white light passes through a biosensor, the interference pattern of the reflected light is dependent on the distance traveled by the beam. When a molecule binds to the surface, a wavelength shift in the optical interference (Δλ) is measured in real-time by the BLI instrument. This signal is used to report the association or dissociation of molecular complexes. (**B**) In a typical BLI trajectory, the interference pattern of a streptavidin-coated biosensor was measured in Buffer H to establish a baseline optical signal (phase 1). Biotinylated human telomeric G4 DNA was loaded onto the senor tip through biotin-streptavidin interactions and an increase in Δλ was measured (phase 2). The biosensor was washed with buffer to confirm the DNA substrate remained surface-immobilized (phase 3). Freely diffusing FANCJ AKKQ peptide was introduced to initiate the association reaction (phase 4). Once a binding equilibrium was reached, the biosensor was placed in buffer to examine the dissociation of AKKQ-G4 complexes. (**C**) BLI experiments were performed in triplicate as a function of FANCJ AKKQ peptide concentration (64, 32, 16, 8, 4, 2, 1, and 0.5 µM). Time-courses were truncated to show the binding and dissociation phases. (**D**) The equilibrium Δλ values at the end of the association phase were plotted against AKKQ concentration to construct a binding isotherm. Fitting this curve to a one-site interaction model yielded K = 9.4 ± 1.1 × 10 ^4^ M ^−1^. BLI experiments with an 8oxo1 or 8oxo5 modification within (TTAGGG)_4_ resulted in small increases in affinity, with K_8oxo1_ = 2.3 ± 0.8 × 10^5^ M^−1^ and K_8oxo5_ = 1.8 ± 0.2 × 10^5^ M^−1^.

**Figure 4 genes-11-00005-f004:**
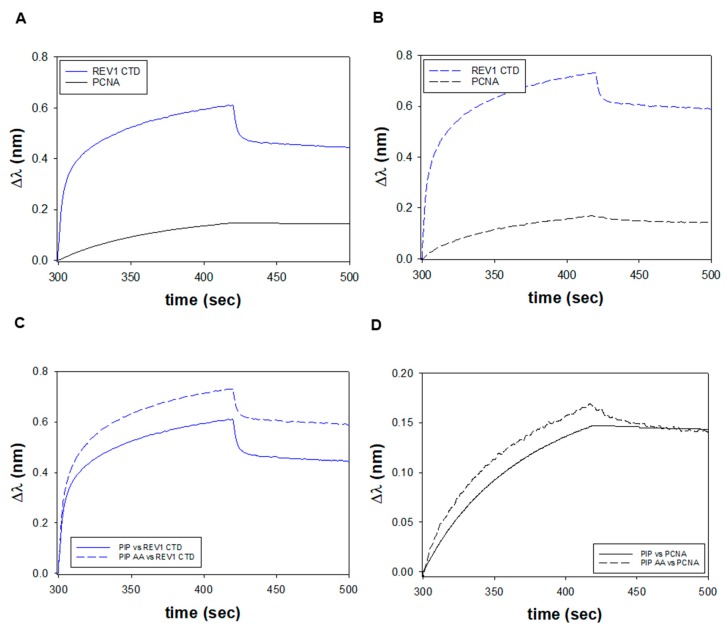
(**A**) BLI time-courses of a biotinylated FANCJ PIP (150 nM), binding to either 1.8 µM REV1 CTD (solid blue) or 1.3 µM PCNA trimer (solid black). (**B**) BLI experiments with biotinylated FANCJ PIP AA (150 nM) and REV1 CTD (dashed blue) versus PCNA (dashed black). (**C**) Overlay of the data from panels A and B for REV1 CTD with FANCJ PIP (solid blue) or PIP AA (dashed blue). (**D**) Overlay of the data from panels A and B for PCNA with the wild-type (solid black) versus the mutant FANCJ peptide (dashed black).

**Figure 5 genes-11-00005-f005:**
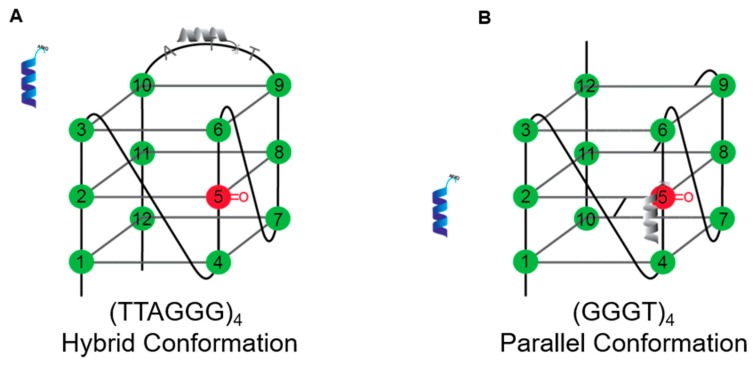
Model for 8oxoG4 recognition by FANCJ AKKQ. (**A**) The human telomeric G4 DNA (TTAGGG)_4_ adopts a hybrid configuration. The FANCJ AKKQ peptide binds with high affinity to the TTA loop and does not interact with an 8oxoG-modified base. Partial unfolding or rearrangement of this structure would therefore have no impact on the AKKQ-G4 interaction. (**B**) (GGGT)_4_ folds into a parallel G4. The shortened loops reduced its affinity for FANCJ AKKQ, which enabled the peptide to interact directly with a surface-exposed 8oxoG.

**Table 1 genes-11-00005-t001:** Peptide sequences and DNA sequences.

**Peptide Sequences**			
Name	Sequence (N→C)	Molar Extinction Coefficient (M^−1^cm^−1^)	Purpose
FANCJ AKKQ-W	SPEKTTLAAKLSAKKQASIW	5500	Fluorescence and BLI
bioFANCJ PIP	Biotin-SWSSFNSLGQYFTGKIP	6990	BLI
bioFANCJ PIP AA	Biotin-SWSSFNSLGQAATGKIP	5500	BLI
**DNA Sequences**			
Name	Sequence (5’→3’)	Molar Extinction Coefficient (M^−1^cm^−1^)	Purpose
(TTAGGG)_4_	TTAGGGTTAGGGTTAGGGTTAGGG	244,600	Fluorescence and CD
(TTAGGG)_4_ 8oxo1	TTA/i8oxodG/GGTTAGGGTTAGGGTTAGGG	239,000	Fluorescence and CD
(TTAGGG)_4_ 8oxo5	TTAGGGTTAG/i8oxodG/GTTAGGGTTAGGG	239,000	Fluorescence and CD
(GGGT)_4_	GGGTGGGTGGGTGGGT	157,200	Fluorescence and CD
(GGGT)_4_ 8oxo1	T/i8oxodG/GGTGGGTGGGTGGGT	159,100	Fluorescence and CD
(GGGT)_4_ 8oxo5	GGGTG/i8oxodG/GTGGGTGGGT	151,600	Fluorescence and CD
5’bioPEG12-(TTAGGG)_4_	/5Biosg//iSp18//iSp18/TTAGGGTTAGGGTTAGGTTAGGG	244,600	BLI
5’bioPEG12-(TTAGGG)_4_-8oxo1	/5Biosg//iSp18//iSp18/TTA/i8oxodG/GGTTAGGGTTAGGGTTAGGG	239,000	BLI
5’bioPEG12-(TTAGGG)_4_-8oxo5	/5Biosg//iSp18//iSp18/TTAGGGTTAG/i8oxodG/GTTAGGGTTAGGG	239,000	BLI

**Table 2 genes-11-00005-t002:** Summary of FANCJ AKKQ-G4 binding results by fluorescence spectroscopy.

DNA Sequence	*K* (M^−1^)	A	*R* ^2^
(TTAGGG)_4_	1.6 ± 0.7 × 10^6^	0.88 ± 0.05	0.9961
(TTAGGG)_4_ 8oxo1	1.3 ± 0.4 × 10^6^	0.90 ± 0.04	0.9974
(TTAGGG)_4_ 8oxo5	1.4 ± 0.5 × 10^6^	0.91 ± 0.05	0.9967
(GGGT)_4_	2.7 ± 0.4 × 10^5^	0.93 ± 0.04	0.9977
(GGGT)_4_ 8oxo1	7.7 ± 0.7 × 10^5^	0.93 ± 0.01	0.9988
(GGGT)_4_ 8oxo5	4.4 ± 0.6 × 10^5^	0.91 ± 0.03	0.9972

**Table 3 genes-11-00005-t003:** Summary of AKKQ-G4 binding results by biolayer interferometry.

DNA Sequence	K (M^−1^)	A	*R* ^2^
bioPEG12-(TTAGGG)_4_	9.4 ± 1.1 × 10^4^	0.74 ± 0.03	0.9973
bioPEG12-(TTAGGG)_4_-8oxo1	2.3 ± 0.8 × 10^5^	0.49 ± 0.06	0.9807
bioPEG12-(TTAGGG)_4_-8oxo5	1.8 ± 0.2 × 10^5^	0.69 ± 0.03	0.9954

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
