# Peer review of "Assembly of a G-Quadruplex Repair Complex by the FANCJ DNA Helicase and the REV1 Polymerase"

_genes, 2019, doi:10.3390/genes11010005_

Round 1
Reviewer 1 Report
This manuscript by Wu and co-workers studies the ability, via fluorescence titration and biolayer interferometry, of FANCJ DNA helicase to recognize potential 8-oxoguanine lesion sites in G-quadruplex DNA (hybrid telomeric G4 and simpler parallel G4 lacking the TTA loop investigated herein) and ability to recruit repair enzymes such as REV1 polymerase to the sites. The topic is novel and of high interest in the field of G-quadruplexes, in light of their potential biological functions and how these can be manipulated pharmaceutically. The manuscript itself is very well written and sufficient bibliographic references are included.
However, a number of critical issues pertaining to the experimental evidence provided need to be addressed in order for the manuscript to become appropriate for publication. This would require major revision.
The Experimental Section does not provide details of concentrations and volumes of DNA and peptide (stock and working) solutions used in the experiments. The pH of the buffer used in CD is also missing from the corresponding section. It is also not specified whether the DNA sequences were annealed and re-folded prior to the experiments (this is important to obtain a well-folded structure, especially given the dynamic nature of the telomeric sequence). In the fluorescence experiment, point (0,0) should be included and "locked" for fitting purposes. Also, I would expect to see some more points in the plateau region of the curves. However, the fit of the model looks fine based on R-squared values. The authors do not attempt to rationalize the inconsistency between fluorescence and biolayer interferometry (1 order of magnitude). I feel this is a result of a poor choice of fluorescent probe. The tryptophan is non-selective as G4 probe, plus it absorbs close to the DNA region, and the combination of these might lead to an overestimation of K values. I would consider the bilayer interferometry to be more accurate on this occasion. For this reason, it would strengthen the manuscript to add one more direct method (e.g. Isothermal Titration Calorimetry, ITC) to back up the findings of the BLI. The claim by the authors for 2 alternative binding sites on G4s, depending on presence or absence of the TTA loop, requires further evidence. I would expect the authors to provide control experiments with the TTA sequence alone to indicate recognition, and perhaps the GGGT sequence in low-salt (unfolded) conditions to obtain indications whether it is the actual primary sequences or their 3D folds that get recognized. Finally the claim for lesion recognition and possible recruitment of repair enzymes does not appear to be general based on the small number of sequences investigated. Perhaps the title and claim should be made more specific to the studied G4s.
Author Response
Reviewer 1:
Thank you for providing us with your valuable and critical review. Please see below for our responses to your concerns.
1) “The Experiment Section does not provide concentrations and volumes of DNA and peptide (stock and working) solutions used in the experiments.” The journal and grant funding agency require all raw and processed data to be made freely accessible. This includes all stock and working concentrations of reagents and volumes of additions in the titrations. All notebook and spreadsheet files will be available on the Open Science Framework data repository as well. The total DNA and peptide concentrations used in the fluorescence and BLI experiments are shown in the binding curves, and since all data will also be made available on OSF, we did not include them again as supplementary tables.
2) “The pH of the buffer used in CD is also missing.” Thank you for catching this oversight. This has been updated in the Materials and Methods. All experiments were performed at pH 7.5.
3) “It is not specified whether the DNA sequences were annealed and re-folded prior to the experiments.” The Materials and Methods section has been revised accordingly. G4 substrates were heated to 95ËšC for 10 minutes and then slowly cooled to 25ËšC prior to the experiments.
4) “In the fluorescence experiment, point (0,0) should be included and ‘locked’ for fitting purposes.” We have analyzed the binding curves with and without inclusion of the origin and it made no impact on the fitted parameters. We agree that point (0,0) should be included in analysis and is a real data point; however, we cannot show the data on a log scale when including x = 0. Because it is more informative to represent binding isotherms on a log scale rather than on a linear scale (as that can be misleading), the origin was omitted from the plots.
5) “I would expect to see more points in the plateau region of the curves. However, the fit of the model looks fine based on R-squared values.” We came to the same conclusion. Based on the R-squared values, we determined that the data was well-described by the model and did not go higher in concentration. Ideally, having more points in the plateau region would be useful but based on the binding constant, this would require millimolar quantities of our samples which was not feasible.
6) “The authors do not attempt to rationalize the inconsistency between fluorescence and BLI (1 order of magnitude).” We have commented on this in the Discussion. While we have not fully understand the discrepancies, our rationale is that in BLI, one of the molecules must be surface-immobilized. The site of attachment and surface orientation of the molecule may therefore occlude interactions with the binding partner to result in lower affinity values. Hence, in our BLI results, we have focused on the relative changes in affinities rather than absolute equilibrium constant values. In our ongoing BLI studies, we are testing the effects of linker length, site of biotinylation, etc, to further dissect the cause of these differences.
7) “I feel this is a result of a poor choice of fluorescence probe. Tryptophan is non-selective as a G4 probe, plus it absorbs close to the DNA region, and the combination of these might lead to an overestimation of K values. I would consider the BLI to be more accurate on this occasion.” Tryptophan was chosen as a probe because the FANCJ AKKQ peptide sequence already has a tyrosine at that position; a tyrosine to tryptophan substitution is relatively modest compared to externally labelling the peptide or the DNA. Intrinsic trytophan fluorescence quenching is a well-established procedure that has been used to examine many protein-DNA interactions. Because these experiments are performed with peptide and DNA free in solution, we actually consider the K values determined from the fluorescence studies to be more reliable than BLI. We used BLI as a secondary method to examine binding because it has higher throughput and requires much less sample material, but BLI measurements must be made with one species placed on the biosensor. Hence, each technique has its distinctive advantages and drawbacks. Although tryptophan fluorescence quenching is robust, it is not suitable for monitoring protein-protein interactions because the signal changes can originate from either species (or both), but this is not a limitation with BLI.
8) “It would strengthen the manuscript to add one more direct method (e.g. ITC) to back up the findings of BLI.” This is an excellent suggestion and we are currently exploring ITC experiments in collaboration with a colleague in the Department of Chemistry here at Oakland University. The changes in binding enthalpy are very small for both the peptide-G4 and peptide-PCNA (REV1) interactions, leading to poor signal to noise. We are looking for a collaborator with a nano-ITC instrument so that we can use higher concentrations of materials at lower volumes to obtain better quality data, but currently our BLI results have been more reliable than ITC.
9) “The claim by the authors for 2 alternative binding sites on G4s depending on presence or absence of the TTA loop requires further evidence. I would expect the authors to provide control experiments with TTA sequence alone to indicate recognition.” This is a working model we have put forth based on the experimental evidence presented herein. We are testing its rigor further in ongoing studies by systematically varying the loop length and composition of the G4s. While we have not done a binding experiment with just TTA or poly(TTA), we have examined FANCJ AKKQ binding to a dT24 substrate since it is devoid of G4s or other secondary structure. We observed reduced affinity, suggesting that the peptide can bind to ssDNA regions. Our current effort to dissect this model involves a comprehensive CD analysis of the G4 substrates as a function of sequence, temperature, solution conditions (salt concentrations and salt type), and 8oxoG positions to determine the thermal stability and conformation of these G4 structures. Once that is established, we will repeat the binding experiments and correlate the observed affinities with the CD data.
10) “…perhaps the GGGT sequence in low-salt (unfolded) conditions to obtain indications whether it is the actual primary sequences of their 3D folds that get recognized.” This is an excellent suggestion by the reviewer and as stated in the point above, we are carrying out these exact experiments so that we can better relate the 3D conformations of the G4s with peptide binding. We also plan on scrambling the G4 sequences and testing their effects on binding.
11) “The claim for lesion recognition and possible recruitment of repair enzymes does not appear to be general based on the small number of sequences investigated.” G4-forming DNA sequences are ubiquitous and diverse. While we have only examined a limited number of (un)damaged G4 substrates in this work, others have shown that without FANCJ, DNA replication stalls precisely at G4-containing regions so its activity can be generalized to G4s. The molecular recognition of damaged G4s by repair proteins such as FANCJ remains an active area of investigation. In this work, we showed that the G4 binding site identified in our previous study also binds to 8oxoG4s and targets G4s independently. Our future work will examine the effects of sequence composition, loop length, damage position, and solution conditions on these interactions to gain additional insight to the structural determinants of 8oxoG4 recognition.
Reviewer 2 Report
This is an extremely well presented manuscript. I greatly enjoyed it.
The introduction did a great job introducing G4 structures and their importance to biological processes.
The methods and materials section is well detailed. The results are well presented and the additional results in the SI enhance the presentation.
The discussion section is also very well presented. I have no suggestions for enhancing the presentation or improving the scientific soundness.
Author Response
Reviewer 2:
Thank you for your comments!
Reviewer 3 Report
Page 1, Lane 40= "A growing number of helicases and polymerases have emerged as key enzymes that unfold G4s in human cells [17-21]." More recent studies can be cited such as PMID:31744872,PMID:31740492, PMID:31548374.
Page 3, Lane 103= Do the authors know the conditions used to lyophilize the peptides? Did they lyophilized in water or any specific buffer?
Page 3, Lane 110= How long DNA substrates were stored at 4 Celsius? Any quality check experiment (for aggregation or higher order structure formation) were done for the DNA sequences before they were used in specific assay?
Page 3, Lane 115= It would be helpful for the reader if the authors name the sequences in table the same with the Figure legends (Figures 1, 2, and Supplementary Figures).
Page 4, Lane 118= It seems both PCNA and REV1 CTD contain Histidine tag. Was it cleaved for binding studies? if not, do authors know whether His-tag has any effect on binding to FANCJ peptide? is it at N-terminus or C-terminus? How many Histidine tags are used?
Page 4, Lane 139 and 140= "The column was washed with 200 mL of 20 mM NaPi (pH 7.5), 30 mM NaCl, 30 mM imidazole, 1 mM DTT, and 5% (v/v) glycerol. PCNA or REV1 CTD was eluted with 20 mM NaPi (pH 7.5), 30 mM NaCl, 1 M imidazole, 1 mM DTT, and 5% (v/v) glycerol over a 200 mL linear gradient." Was the salt concentration 30 or 300 mM?
Page 5, Lane 193= Was binding tested with a negative control sequence (G-rich sequence but unable to form G-quadruplex)?
Page 6, Lane 209= "However, all three DNA sequences adopted distinct structures" seems to be very strong statement since the conclusion is made only based on CD study and the current study does not have any further data provided for structural analysis.
Page 6, Lane 229= There is extra "." at the end of the sentence.
Page 8, Lane 278= Figure (4D) should be Figure (3D)??
Page 10, Lane 345= "We initially predicted that the incorporation of 8oxoG would destabilize G4s and that the AKKQ peptide may have a greater affinity for these structures. To our surprise, the CD measurements and binding data led to a different result. The human telomeric G4 (TTAGGG)4 adopted a hybrid conformation in KCl-containing buffer, and 8oxoG modifications at the 8oxo1 and 8oxo5 positions disrupted G4-folding as anticipated" The CD measurements suggest the folding is effected but there is no evidence for destabilization of the G4s.
Author Response
Reviewer 3:
Thank you for your constructive feedback. Please see below for our responses and how we have addressed these concerns in the main text.
1) “More recent studies can be cited such as…” the Introduction section has been revised accordingly with the updated references. I have kept the older references also for historical purposes but can remove them it the editor feels that is necessary.
2) “Do the authors know the conditions used to lyophilize the peptides?” We have reached out to Genscript regarding this concern. The peptides were cleaved off the column after synthesis using trifluoroacetic acid (TFA). The peptides were then lyophilized in water as a chloride salt.
3) “How long were DNA substrates stored at 4ËšC?” Unlike plasmid DNA, DNA oligos are stable and can be safely stored 4ËšC. They are used up typically within 3-6 months at which point new oligos are purchased from IDT. We have not seen signs of degradation or aggregation by gel electrophoresis and our experiments with different DNA preparations have been reproducible. If issues arise from prolonged storage at 4ËšC, substrates will be aliquoted and frozen so that only one sample is maintained at 4ËšC at a time.
4) “It would be helpful for the reader if the authors name the sequences in the table the same with the Figure Legends.” The table has been revised accordingly to match the substrate names used in the Figures and Figure Legends.
5) “Do authors know whether His-tag has any effect on binding to FANCJ peptide?” Our protein constructs have a His-tag at the N-terminus that can be cleaved by thrombin (PCNA) or TEV (REV1 CTD) protease. We have not removed the His-tag in our experiments because we are currently using Ni-NTA biosensors in our BLI experiments to monitor protein-protein interactions and want to compare the results directly with those from this manuscript. We have not tested the influence of His-tag on FANCJ peptide binding which is always a caveat in working with tagged proteins. Work published using His-tagged REV1 CTD bound to an RIR peptide (Ohashi et al, 2009 Genes Cells) so it is unlikely presence of the tag has adverse effects, but we will test this further for FANCJ peptide binding.
6) “Was the salt concentration 30 or 300 mM?” This has been revised in the Materials and Methods. For initial protein loading and for the first wash, 300 mM salt was used to reduce non-specific binding. The column was then washed with buffer contain 30 mM salt, and the protein was eluted in 30 mM salt over an imidazole gradient.
7) “Was binding tested with a negative control sequence (G-rich but unable to form G4)?” This is an excellent suggestion and we have done something similar. Polynucleotide sequences (with the exception of polythymine) can adopt secondary structures which can be misleading. We have tested binding using a dT24 sequence which cannot form a G4 and is devoid of secondary structure. The FANCJ peptide still bound to this substrate albeit with lower affinity, but this result is consistent with our model that the peptide can bind to the ssDNA region of the TTA loop. We have not tried using randomized G-rich sequences yet but this aligns with our current studies of the (TTAGGG)4 and (GGGT)4 G4s. We plan on incorporating 8oxoG at every guanine position and testing their stability and folding by circular dichroism at various temperature (see also point #11). Using the same substrate composition but with randomized guanine positions would be a good control to include for these experiments.
8) “Seems to be a very strong statement since the conclusion is made only based on CD study and the current study does not have any further data provided for structural analysis.” The text has been revised so that we are not overstating that claim. The reviewer is absolutely correct that CD results alone do not provide any distance information and so the word “structure” should be used carefully. Since we do observe three distinct CD patterns, we concluded that the three DNA sequences adopted different folds (or conformations). They can certainly have similar structural features that can be further examined with other techniques.
9) “Extra ‘.’ at the end of the sentence.” This has been removed.
10) “Figure 4D should be Figure 3D?” Thank you for catching this typo. This has been corrected.
11) “CD measurements suggest folding is effected but there is no evidence for destabilization of the G4s.” The reviewer again is absolutely correct (see point #8) that our current CD results do not provide any information about thermal stability. Destabilization of G4s by 8oxoG at these specific positions have been published by a different group. We cited this work throughout the text and this was our motivation for choosing these damage positions. We hypothesized that G4-destabilization by 8oxoG (observed by others) would weaken peptide binding but saw the opposite result, and so “destabilization” was used in that context alone. We concluded from our CD experiments that 8oxo1 and 8oxo5 disrupted G4-folding but did not mention stability. We are examining the thermal stability of these G4 substrates by repeating CD experiments at different temperature, solution conditions, and DNA damage position so that we can better comment on this in a future publication.
Round 2
Reviewer 1 Report
With the changes done by the authors and the responses provided to my previous comments, I find the manuscript to be now suited.